# Barrelier's Speedwell (*Veronica barrelieri* Schott ex Roem. et Schult., Plantaginaceae)—Potential of Free Volatile Compounds for Horticulture

**Dario Kremer** [1], **Edith Stabentheiner** [2], **Marija Nazlić** [3], **Marko Randić** [4], **Siniša Srečec** [5] **and Valerija Dunkić** [3,*]

1    Faculty of Pharmacy and Biochemistry, University of Zagreb, A. Kovačića 1, 10000 Zagreb, Croatia
2    Institute of Biology, Karl-Franzens University, Schubertstrasse 51, A-8010 Graz, Austria
3    Faculty of Science, University of Split, Ruđera Boškovića 33, 21000 Split, Croatia
4    Public Institution "Priroda", Grivica 4, 51000 Rijeka, Croatia
5    Department of Plant Production, Križevci College of Agriculture, Milislava Demerca 1, 48260 Križevci, Croatia
*    Correspondence: dunkic@pmfst.hr; Tel.: +385-21-619-296

**Abstract:** Barrelier's Speedwell or *Veronica barrelieri* Schott ex Roem. et Schult. (syn. *Pseudolysimachion barrelieri* (Schott ex Roem. et Schult.) Holub (family Plantaginaceae) grows on dry grasslands and rocky slopes in southeastern Europe. Because of its attractive blue flowers arranged in dense inflorescences up to 30 cm long, this plant has great potential for horticulture, especially in dry climates. As part of studies on biologically active compounds in this species, free VCs (Volatile Compounds) were analyzed by GC-MS (Gas Chromatography with Mass Spectrometry) and micromorphological features were studied by SEM (Scanning Electron Microscopy). Free VCs from aboveground plant parts collected during flowering were characterized by a considerably high content of oxygenated diterpene phytol, followed by hexadecanoic acids, pentacosane, and caryophyllene oxide. These compounds are most abundant in the composition of VCs isolates of *V. barrelieri* from all five Croatian localities studied. Non-glandular and two subtypes of capitate glandular trichomes were detected on the stems, leaves and calyx of *V. barrelieri*. *Veronica barrelieri* attracts pollinators with its attractive flower appearance and specialized metabolites such as free VCs, which are environmentally friendly and possible natural botanical pesticides.

**Keywords:** free volatile compounds; GC-MS; phytol; *Pseudolysimachion barrelieri*; speedwell





## 1. Introduction

The subgenus or section *Pseudolysimachion* W. D. J. Koch of the genus *Veronica* L. (family Plantaginaceae) encompass from 10 to about 40 species distributed in Eurasia [1,2]. Due to some traits (long corolla tube, dense inflorescences, 17 chromosomes) some authors consider these species as a separate genus *Pseudolysimachion* (W. D. J. Koch) Opiz [3–5]. The Altai Mountains region with thirteen registered species is the center of sect. *Pseudolysimachion* diversity while the second center is the northern Balkan Peninsula with six species [6]. Five of these species are naturally distributed in Croatia [7]. Among European species, *Veronica spicata* L. (syn. *Pseudolysimachion spicatum* (L.) Opiz) has the widest natural distribution range covering most of Europe [8], as well as Central Asia, western and eastern Siberia [1].

The studied species Barrelier's Speedwell or *Veronica barrelieri* Schott ex Roem. et Schult. (syn. *Pseudolysimachion barrelieri* (Schott ex Roem. et Schult.) Holub, *Veronica spicata* ssp. *barrelieri* (Schott ex Roem. et Schult.) Murb) is distributed in southeastern Europe and extends to northern Italy (Figure 1). It is a perennial herbaceous plant with rhizome and erect or sometimes ascending stem, usually growing up to 35 cm tall, only sometimes higher. The leaves are opposite, linear-lanceolate to ovate, with crenate to subentire margin, up to 8 cm long, subsessile or with short petiole (the petiole of lower leaves is up to 1 cm long). Blue, decorative flowers are arranged in terminal racemes up to 30 cm long

(Figure 2a). The calyx is deeply divided into four, ovate-elliptical, obtuse, ciliate segments. Campanulate corolla is up to 8 mm wide, with tube longer than wide [8]. The typical habitat of Barrelier's Speedwell are nutrient-poor, dry grasslands and meadows, rocky slopes, shrubbery and roadsides, in open or eventually semi-shady places, from sea level to subalpine vegetation belt [8,9].

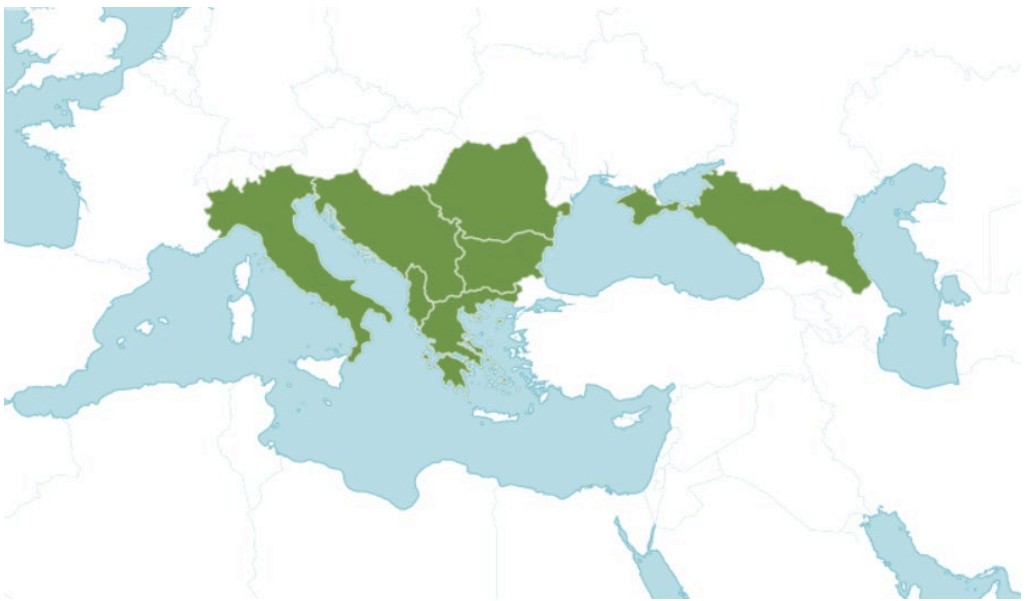

**Figure 1.** Distribution of Barrelier's Speedwell. Source: https://powo.science.kew.org/taxon/urn:lsid:ipni.org:names:811672-1 (accessed on 10 July 2022).

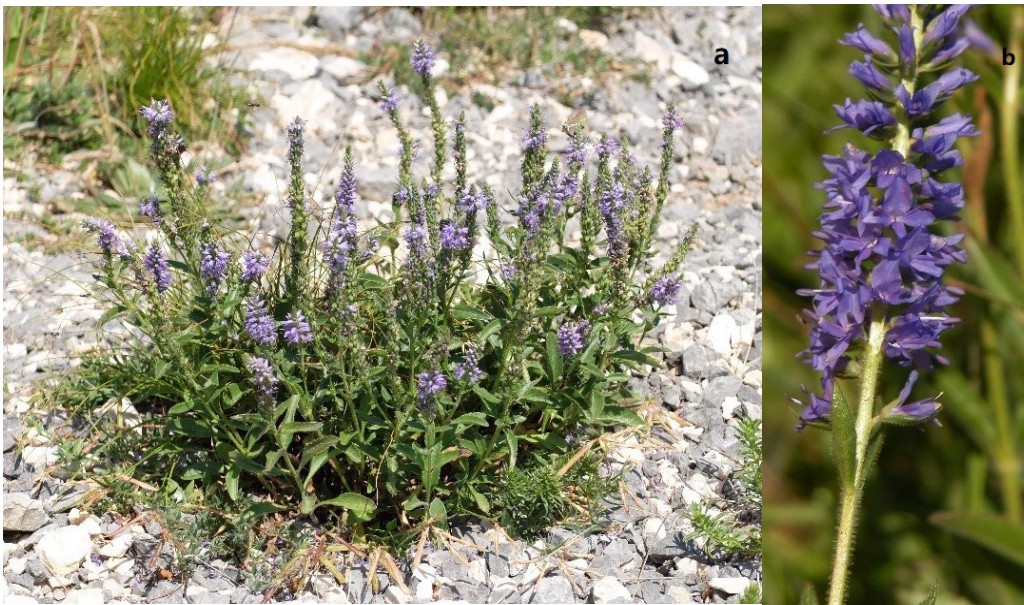

**Figure 2.** Barrelier's Speedwell in its natural habitat (**a**) and her inflorescence (**b**).

As a small, decorative, non-invasive species adapted to dry habitats Barrelier's Speedwell is suitable for planting in rock gardens, especially in Mediterranean areas with a constant water shortage (Figure 2). The problem of water scarcity is expected to worsen in the future due to impending climate changes. Therefore, the demand for drought-tolerant ornamental plant species is likely to increase in the future. Also, Barrelier's Speedwell could potentially be a medicinal plant species because the closely related species *V. spicata* is used in traditional medicine of Balkan people as a tea for treatment cough and throat

rinsing [10]. In addition, Lee et al. [11] found that iridoid glycoside verproside isolated from species belonging to the sect. *Pseudolysimachion* is the most effective anti-asthmatic agents among the investigated glycosides. Many other closely related *Veronica* species are also used for the treatment different ailments in traditional medicines of many nations around the world [12–15]. Some *Veronica* species from the sect. *Pseudolysimachion* are also used in horticulture, like cultivars of *V. spicata* and *V. longifolia* L. (syn. *Pseudolysimachion longifolium* (L.) Opiz) [2,16].

The free VCs (Volatile Compounds) are one of the plant products that can possess strong biological activity. In addition to the effect of all components, biological activity can also be influenced by individual components in the composition of the isolate. Thus, in the composition of the studied isolate of *V. barellieri*, the main component is phytol, a diterpene alcohol from chlorophyll known for its medicinal properties. Phytol has antimicrobial, antioxidant and anti-inflammatory activity. Several studies have demonstrated the anticancer effects of plant extracts with phytol as the main ingredient. The effects of phytol were also confirmed on apoptosis in hepatocellular carcinoma cells and the cytotoxic effect of phytol on cell lines in vitro [17]. The content of VCs in closely related species *Veronica linariifolia* Pall. ex Link (syn. *Pseudolysimachion linariifolium* (Pall. ex Link) Holub), *V. longifolia* L. (syn. *Pseudolysimachion longifolium* (L.) Opiz), and *V. spicata* were investigated by Feng et al. [18], Osmachko et al. [19], and Dunkić et el. [20], respectively. To the best of our knowledge there is no information about free VCs of Barrelier's Speedwell. As a first step in the study of this species an analysis of VCs was conducted. To get insight in places of VCs production and storage the micromorphological traits of glandular trichomes were also studied. Thus, the aim of this paper is to obtain GC-MS profile of free VCs of Barrelier's Speedwell and to learn the anatomical structure of the glandular hairs where VCs are formed, in order to discover the biological potential for the horticultural development of this ornamental species.

## 2. Materials and Methods

### 2.1. Plant Material

Collection of plant material of Barrelier's Speedwell was performed during the flowering period at five locations in Croatia (Table 1, Figure 3). The voucher specimens were stored in the "Fran Kušan" Herbarium (HFK-HR), University of Zagreb, Faculty of Pharmacy and Biochemistry, Zagreb, Croatia. The above-ground part of the plant, i.e., flowers, leaves and stems, for GC (Gas Chromatography) and GC-MS (Gas Chromatography with Mass Spectrometry) analyses were air dried at room temperature (22 °C) and sheltered from direct sunlight for 15 days. After drying, plant material was stored in a double paper bags and kept in the dark. Plant material for SEM (Scanning Electron Microscopy) study was collected from ten plants per site and placed in solution mixture prepared from 96% ethanol, water, formalin, and acetic acid in a volume ratio of 70:20:5:5. After four days, the plant material was transferred to 70% ethanol and placed in the refrigerator at 4 °C.

**Table 1.** Locations of plant material of Barrelier's Speedwell collection.

| Locality | Latitude | Longitude | Altitude a.s.l. (m) | Voucher No. |
|---|---|---|---|---|
| 1. Lič Polje | 45°16′27″ N | 14°45′00″ E | 728 | HFK-HR-2021-35 |
| 2. Velebit Mt | 44°43′56″ N | 14°56′57″ E | 1162 | HFK-HR-2021-49 |
| 3. Gornja Kamenica * | 45°02′43″ N | 15°07′41″ E | 645 | HFK-HR-2021-44 |
| 4. Dinara Mt | 44°01′54″ N | 16°24′36″ E | 1198 | HFK-HR-2021-72 |
| 5. Kamešnica Mt | 43°42′07″ N | 16°48′16″ E | 1250 | HFK-HR-2021-81 |

* = sample for pollen analysis.

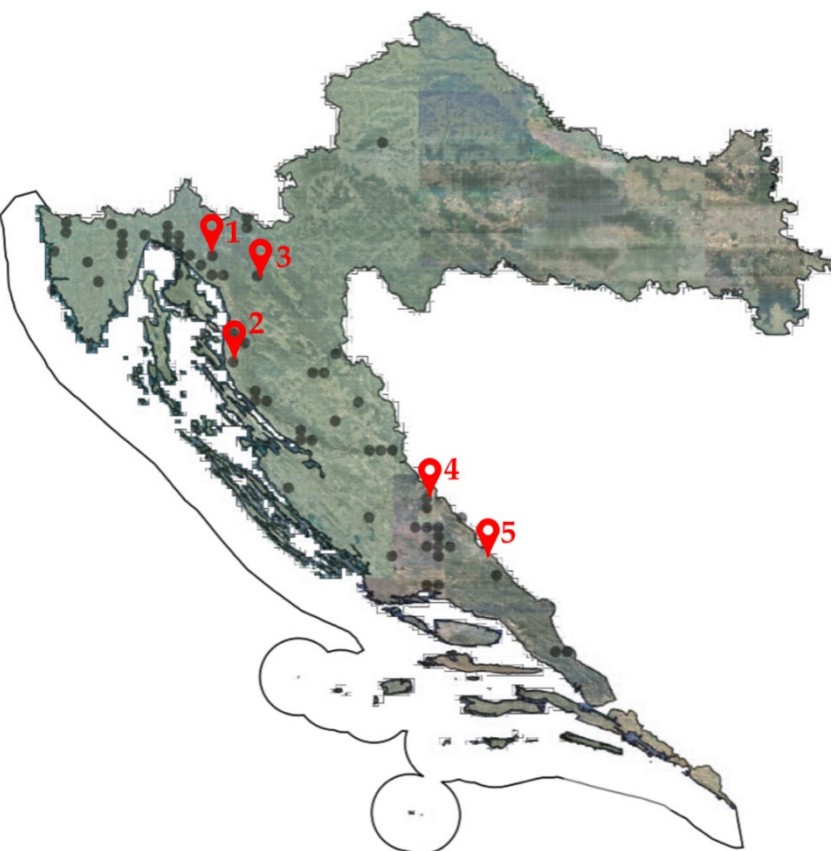

**Figure 3.** Distribution of Barrelier's Speedwell in Croatia (black dots) and locations where plant material was collected from for this research (red marks, numbers 1–5 stated in the Table 1). Source for the map with a distribution of the studied species in Croatia: Nikolić T. ur. (2015—onwards): Distribution of *Pseudolysimachion barrelieri* (Schott ex Roem. et Schult.) Holub in Croatia, Flora Croatica Database (http://hirc.botanic.hr/fcd). Faculty of Science, University of Zagreb, accessed on: 10 July 2022. On this basic map authors added red marks of the studied locations.

*2.2. GC-MS Analyses*

Fifty grams of each plant sample was hydrodistilled for 3 h in a Clevenger apparatus and free VCs obtained were collected in the penthan/diethileter mixture. Then, the VCs were analyzed in detail using GC-MS analytical methods, GC model 3900 (Varian Inc., Lake Forest, CA, USA) and MS (model 2100T; Varian Inc. Lake Forest, CA, USA). The columns used are non-polar CP (Capillary Column)VF-5ms (30 m × 0.25 mm inside diameter, coating thickness 0.25 μm, Palo Alto, CA, USA) and polar capillary column CPWax 52 CB (30 m × 0.25 mm i.d., coating thickness 0.25 μm, Palo Alto, CA, USA). The application method is the following and described in detail in Nazlić et al. [21]: the gas carrier was helium at 1 mL min$^{-1}$. The conditions for the polar column were: temperature 0 °C isothermal for 3 min, and then increased to 246 °C at a rate of 3 °C min$^{-1}$, and held isothermal for 25 min. Conditions for the CP Wax 52 column were: temperature 70 °C isothermal for 5 min, and then increased to 240 °C at a rate of 3 °C min$^{-1}$ and held isothermal for 25 min. The injected volume was 2 μL and the split ratio was 1:20. The MS Identification (triplicate analyses) of individual peaks was performed by comparing their retention indices of n-alkanes with literature data and authentic standards [22].

*2.3. Micromorphological Analyses*

The whole calyx, the middle part of the leaves and the internode of stem were used to prepare the samples for micromorphological studies. The application method is described in detail by Kremer et al. [23]. Pollen from several flowers per each of totally ten plants

collected at locality Gornja Kamenica was taken out from anthers after critical point drying. Then, pollen was mixed to obtain random sample. The length of 30 pollen grains was measured and expressed as mean ± standard deviation (SD). The observed trichomes and pollen type were described using standard terminology [24,25].

### 2.4. Statistical Analyses

Statistical analysis was performed in GraphPad Prism Version 9 (Dotmatics, 2365 Northside Dr., Suite 560, San Diego, CA, USA). All data are expressed as mean ± SD ($n \geq 3$). In order to summarize the data, the values of essential oil volatiles in the amount larger than 1% from all five locations were subjected to PCA (Principal Component Analysis PCA). The purpose of conducting PCA analysis is to objectively determine which populations of *Veronica barrelieri* are more similar to each other in terms of GC-MS profile.

## 3. Results and Discussion

### 3.1. GC-MS Analyses

The chemical components determined in the VCs of Barrelier's Speedwell from five Croatian localities are listed in Table 2. The yield is 0.07% ($w/w$) for all isolates.

**Table 2.** Chemical composition of the volatiles of Barrelier's Speedwell from five different locations.

| Locality | | | Lič Polje | Gornja Kamenica | Velebit Mt | Dinara Mt | Kamešnica Mt |
|---|---|---|---|---|---|---|---|
| Component | RIa | RIb | VC ± SD | VC ± SD | VC ± SD | VC ± SD | VC ± SD |
| α-Thujene | 924 | 1012 | 0.82 ± 0.01 | 0.63 ± 0.01 | 1.15 ± 0.01 | | 0.64 ± 0.01 |
| α-Pinene * | 935 | 1017 | – | – | – | – | 0.70 ± 0.01 |
| Benzaldehyde | 952 | 1508 | 2.17 ± 0.01 | 0.533 ± 0.01 | 0.97 ± 0.01 | 1.25 ± 0.01 | 3.98 ± 0.01 |
| β-Phellandrene | 1002 | 1195 | – | – | – | 0.75 ± 0.1 | – |
| Benzene acetaldehyde | 1036 | 1633 | 3.06 ± 0.01 | 0.56 ± 0.01 | 1.33 ± 0.01 | – | 1.12 ± 0.01 |
| γ-Terpinene | 1057 | 1225 | – | – | – | 1.64 ± 0.01 | – |
| Linalool | 1095 | 1506 | 2.26 ± 0.01 | 0.70 ± 0.01 | 1.86 ± 0.01 | 2.32 ± 0.01 | 1.53 ± 0.15 |
| n-Nonanal | 1100 | 1389 | – | – | 1.69 ± 0.02 | – | 1.68 ± 0.01 |
| Terpinen-4-ol | 1174 | 1686 | 0.54 ± 0.01 | – | 1.12 ± 0.01 | – | – |
| Borneol | 1176 | 1719 | – | – | – | 1.59 ± 0.01 | – |
| α-Terpineol | 1184 | 1660 | 2.95 ± 0.01 | 3.31 ± 0.01 | 0.66 ± 0.01 | 3.08 ± 0.03 | 1.31 ± 0.01 |
| trans-p-Mentha-1(7),8-dien-2-ol | 1187 | 1803 | – | – | 0.72 ± 0.01 | – | 0.79 ± 0.01 |
| α-Copaene | 1377 | 1484 | – | – | – | – | 0.55 ± 0.01 |
| (E)-β-Damascenone | 1384 | 1819 | 3.86 ± 0.01 | 1.42 ± 0.01 | 0.42 ± 0.01 | 5.69 ± 0.01 | – |
| Methyl eugenol | 1403 | 2005 | 1.01 ± 0.01 | 0.33 ± 0.01 | 1.85 ± 0.01 | 2.34 ± 0.01 | 2.21 ± 0.01 |
| E-Caryophyllene * | 1424 | 1585 | 1.23 ± 0.01 | 2.03 ± 0.01 | 1.43 ± 0.1 | 3.48 ± 0.01 | 1.48 ± 0.01 |
| (Z)-Methyl isoeugenol | 1451 | 2070 | – | 1.35 ± 0.01 | – | – | – |
| allo-Aromadendrene | 1465 | 1662 | – | – | 0.97 ± 0.01 | 2.59 ± 0.01 | 1.21 ± 0.01 |
| β-Chamigrene | 1478 | 1724 | – | – | – | 0.67 ± 0.02 | – |
| Germacrene D | 1481 | 1692 | 0.48 ± 0.01 | 0.71 ± 0.01 | 0.56 ± 0.01 | 2.43 ± 0.01 | 2.61 ± 0.01 |
| β-Ionone | 1487 | 1935 | 4.34 ± 0.01 | 4.11 ± 0.01 | 3.56 ± 0.01 | 3.42 ± 0.01 | 3.15 ± 0.01 |
| δ-Selinene | 1492 | 1756 | 0.66 ± 0.01 | – | – | 0.54 ± 0.01 | – |
| δ-Cadinene | 1517 | 1745 | – | – | – | – | 0.81 ± 0.01 |
| Spathulenol | 1577 | 2101 | – | – | 0.45 ± 0.01 | 0.25 ± 0.01 | 0.83 ± 0.01 |
| Caryophyllene oxide * | 1581 | 1955 | 4.43 ± 0.01 | 4.18 ± 0.01 | 3.43 ± 0.01 | 3.76 ± 0.01 | 4.51 ± 0.01 |
| Viridiflorol | 1592 | 2099 | 0.65 ± 0.01 | – | – | – | 0.54 ± 0.01 |
| γ-Eudesmol | 1632 | 2175 | – | – | – | – | 0.22 ± 0.01 |
| α-Muurolol | 1645 | 2181 | 2.38 ± 0.01 | 2.06 ± 0.01 | 1.33 ± 0.01 | 2.31 ± 0.01 | 2.52 ± 0.01 |
| Hexahydrofarnesyl acetone * | 1839 | 2113 | 1.56 ± 0.01 | 1.14 ± 0.01 | 0.56 ± 0.01 | 1.31 ± 0.1 | 0.84 ± 0.01 |
| Phytol * | 1942 | 2610 | 37.78 ± 0.01 | 40.32 ± 0.1 | 30.08 ± 0.1 | 32.88 ± 0.02 | 36.43 ± 0.01 |
| Hexadecanoic acid * | 1959 | 2912 | 12.45 ± 0.01 | 12.14 ± 0.1 | 8.75 ± 0.01 | 9.04 ± 0.01 | 12.44 ± 0.01 |
| Eicosane * | 2000 | 2000 | 0.38 ± 0.0 | 3.36 ± 0.01 | 1.22 ± 0.01 | - | 1.12 ± 0.01 |
| Heneicosane * | 2100 | 2100 | – | 0.65 ± 0.01 | 1.98 ± 0.1 | 1.32 ± 0.01 | – |
| Docosane * | 2200 | 2200 | 2.01 ± 0.01 | 7.69 ± 0.01 | 2.13 ± 0.01 | 2.15 ± 0.01 | 0.43 ± 0.01 |
| Tricosane * | 2300 | 2300 | – | – | – | 1.66 ± 0.01 | – |
| Tetracosane * | 2400 | 2400 | 4.61 ± 0.01 | 3.38 ± 0.01 | 9.32 ± 0.01 | 1.72 ± 0.01 | – |
| Pentacosane * | 2500 | 2500 | 3.54 ± 0.01 | 4.03 ± 0.01 | 10.71 ± 0.1 | 5.25 ± 0.1 | 10.32 ± 0.01 |
| Octacosane * | 2800 | 2800 | 0.75 ± 0.01 | – | 6.10 ± 0.01 | – | – |

**Table 2.** *Cont.*

| Locality | | | Lič Polje | Gornja Kamenica | Velebit Mt | Dinara Mt | Kamešnica Mt |
|---|---|---|---|---|---|---|---|
| **Component** | **RIa** | **RIb** | **VC ± SD** | **VC ± SD** | **VC ± SD** | **VC ± SD** | **VC ± SD** |
| Total identification (%) | | | 93.92 | 94.63 | 94.35 | 93.41 | 93.97 |
| Grouped compounds (%) | | | | | | | |
| Monoterpene hydrocarbons | | | 0.82 | 0.63 | 1.15 | 0.75 | 1.34 |
| Oxygenated monoterpenes | | | 5.75 | 4.01 | 4.36 | 8.63 | 3.63 |
| Sesquiterpene hydrocarbons | | | 2.37 | 2.74 | 2.96 | 9.68 | 6.66 |
| Oxygenated sesquiterpenes | | | 9.02 | 7.38 | 5.77 | 7.63 | 9.46 |
| Oxygenated diterpenes | | | 37.78 | 40.32 | 30.08 | 32.88 | 36.43 |
| Hydrocarbons | | | 11.29 | 19.11 | 31.46 | 12.1 | 11.87 |
| Others | | | 26.89 | 20.44 | 18.57 | 21.74 | 24.58 |

Retention indices (RIs) were determined relative to a series of n-alkanes (C8–C40) on capillary columns VF5-ms (RIa) and CPWax 52 (RIb); Identification method: RI, comparison of RIs with those in a self-generated library reported in the literature [22] and/or with authentic samples; comparison of mass spectra with those in the NIST02 and Wiley 9 mass spectral libraries; * injection of reference compounds; –, not identified; SD, standard deviation of triplicate analysis.

The oxygenated diterpene phytol is the most important compound at all localities studied, with an identification percentage of at least 30.08% from the Velebit Mt locality to a maximum of 40.32% from Kamešnica Mt. Benelli et al. [26] studied the activity of phytol, the most abundant compound in the essential oil composition of *Stevia rebaudiana* (Bertoni) Hemsl., and showed strong activity against aphids. Moreover, these researchers suggest that phytol, *(E)*-nerolidol and spathulenol can be used as eco-friendly green insecticides against aphids [26]. To support the production of biopesticides, Pavela et al. [27] again conducted research with exogenously applied phytol, which inhibited the invasion of root-knot nematodes into the roots of *Arabidopsis* Heynh. [28]. Islam et al. [29] have described a wide range of biological activities of phytol and its derivatives in a review article, which gives us further reasons for the horticultural cultivation of Barrelier's Speedwell.

In general, there are only few reports about VCs content in *Veronica* species from sub-genus *Pseudolysimachion*. The analyses of VCs in the closely related species *V. spicata* showed that among 27 identified compounds dominate phytol (21.13%), heptacosane (10.22%), piperitone (9.43%) and pentacosane (8.91%) [20]. Hydrocarbons pentacosane is present in all five investigated localities of Barrelier's Speedwell species with the highest percentage of identification at the Kamešnica Mt locality with 10.32% and in locality Velebit Mt with 10.71% (Table 2). On the other hand, Feng et al. [18] found that the main components of the VCs of *V. linariifolia* were: germacrene D (4.99%), β-myrcene (10.42%), β-phellandrene (10.49%), 1S-α-pinene (10.65%), β-pinene (11.61%), and cyclohexene (25.83%). The major compounds in VCs obtained from leaves of *V. longifolia* were palmitic acid (24.7%), squalene (22.1%), phytol (8.2%), and oleic acid (7.6%), while the major compounds in VCs from flowers were palmitic acid (31.2%), squalene (21.6%), oleic (10.9) and linolenic (10.5%) acids [19]. Also, hexadecanoic acid (palmitic acid) was significantly identified in all five isolates of the studied species Barrelier's Speedwell with isolation percentages from 8.75% from Velebit Mt to 12.45% in locality Lič Polje (Table 2). Looking the general distribution of volatiles by main categories it can be observed that proportions of volatiles are similarly distributed in EOs (Essential Oils) from all five locations (Figure 4). Somewhat higher percentage of VCs from the "Hydrocarbons" category is detected in EO from Velebit Mt location. This EO has high relative percentage of tetracosane (9.32%) and pentacosane (10.71%).

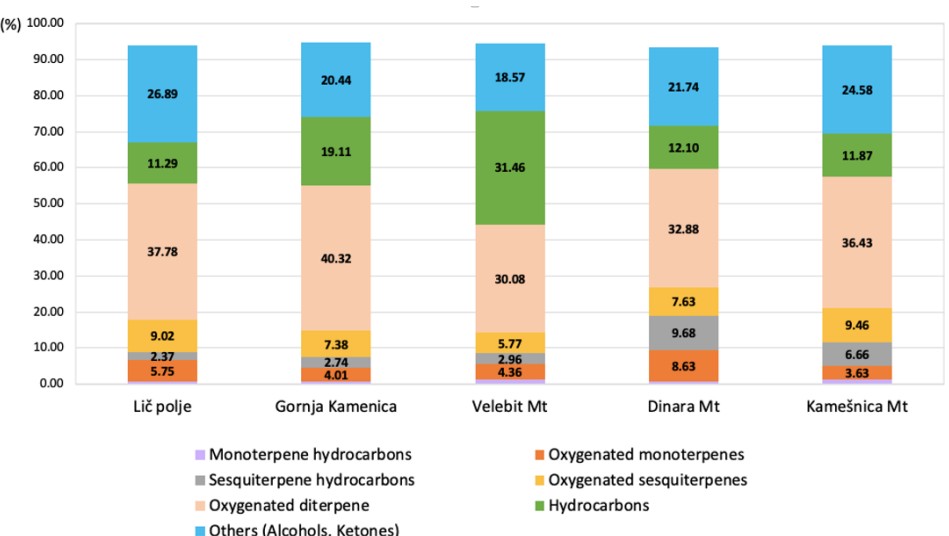

**Figure 4.** Volatile compounds distribution by categories of Barrelier's Speedwell EOs from five locations.

There are more literature sources about VCs content in *Veronica* species from other sections. For example, the main constituents in VCs of *V. thymoides* P. H. Davis ssp. *pseudocinerea* M. A. Fischer from Turkey were hexatriacontene (21.0%), 17-pentatriacontene (8.6%), arachidic acid (8.0%), and α-selinene (7.5%) [13]. Nazlić et al. [21] investigated VCs of *V. saturejoides* Vis. ssp. *saturejoides* from two localities. They found that hexahydrofarnesyl acetone (30.13%), and caryophyllene oxide (20.25%) were the most represented compounds from one locality while hexadecanoic acid (37.31%), and hexahydrofarnesyl acetone (23.24%) were the most abundant from the second [21]. According to Nazlić et al. [30] the major compounds in the lipophilic fraction of *V. officinalis* L. were hexadecanoic acid (20.62%), β-ionone (17.88%), hexahydrofarnesyl acetone (13.92%), and E-caryophyllene (6.78%). Nazlić et al. [31] also investigated free VCs content of *V. austriaca* L. ssp. *jacquinii* (Baumg.) Watzl and found that hexahydrofarnesyl acetone (23.34–52.56%), and hexadecanoic acid (26.71–58.91%) were the most common compounds in all five isolates.

On the other hand, Çelik et al. [32] found that the VCs of *Veronica* sp. contained mainly carvacrol (7.28%), and linalool (4.18%). Finally, Valyova et al. [33] found that ethanol extracts of *V. officinalis* from Bulgaria contained terpinen-4-*ol* (8.96%), as well as phytol, neophytadiene, squalene hexahydrofarnesyl acetone, and vitamin E.

Based on the composition of the free VCs, the species Barrelier's Speedwell can be classified into a specific section, because in addition to the main compounds mentioned above (phytol, hexadecanoic acid, and pentacosane), compounds such as benzaldehyde, linalool, α-terpineol, E-caryophyllene, germacrene D, β-ionone, α-muurolol, hexahydrofarnesyl acetone, methyl eugenol, and docosane were identified in samples from all five localities (Table 2). These identified VCs, as well as some compounds formed by the same metabolic pathway, were previously identified in all *Veronica* species studied and represent an important biological potential for the genus *Veronica* and, consequently, for the studied species Barrelier's Speedwell.

### 3.2. Principal Component Analyses

PCA analyses were performed for VCs of EOs from five locations with the amount greater than 1% (Figure 3). PC1 and PC2 for volatile compounds explained 66.01% of the variance. Two clusters were distinguished (Figure 5a). EOs from the Lič Polje and Gornja Kamenica sites were distinguished as one cluster. The second cluster contains EOs from the Velebit Mt and Kamešnica Mt sites. EO from the Dinara Mt site separated from both clusters as a separate species. The components differentiating EOs VCs in the first cluster are located in the negative region of PC1 and PC2. The major components that

differentiate this cluster are *β*-ionone, phytol, caryophyllene oxide, phytol, hexadecanoic acid and *(Z)*-methyl isoeugenol (Figure 5b). The main components that differentiate second cluster are terpinen-4-*ol*, *n*-nonanal and α-thujene. They are located in the negative region of PC1 and positive region of PC2. The main compounds that separated Dinara Mt EO from both clusters are α-muurolol, *(E)*-*β*-damascenone, *γ*-terpinene, *E*-caryophyllene, borneol and *allo*-aromadendrene (Figure 3b). They are mainly located in the positive region of PC1 and negative region of PC2. In the previous PCA studies on the *Veronica* species VCs it was shown that two species can differentiate into clusters based on the VCs composition [31]. Nazlić et al. showed that *V. saturejoides* ssp. *saturejoides* differentiates from *V. austriaca* ssp. *jacquinii*, especially based on the volatiles extracted in the hydrosols [31].

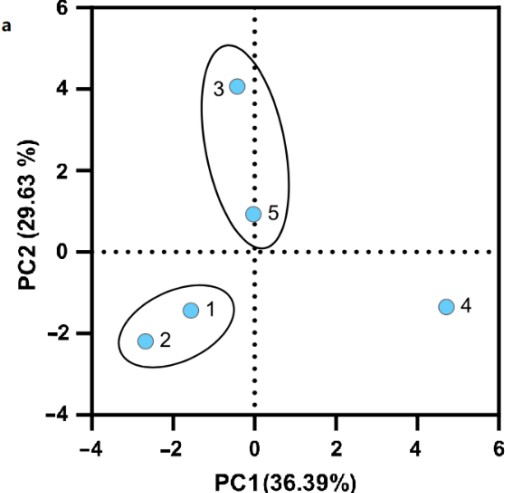

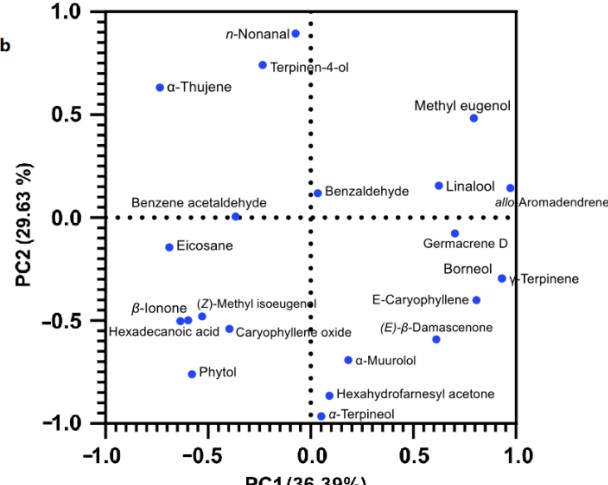

**Figure 5.** PCA analyses of volatile compound in the amount larger than 1% from essential oils (EOs of Barrelier's Speedwell from five locations). (**a**) PCA score plot allocating different locations into clusters (locations numbers 1–5 stated in the Table 1); (**b**) PCA loading plots of volatiles from the first and second principal component.

### 3.3. Micromorphological Studies

The micromorphological studies conducted using SEM revealed the presence of two main trichome types, NG (Non-Glandular) and glandular, on all investigated plant parts (Figure 6). The non-glandular trichomes in *V. barrelieri* are unbranched, acicular trichomes consisting of two (short trichomes) to several (longer trichomes) cells. The short trichomes are upright while long trichomes are folded in different ways. On the calyx, the NG

trichomes are present only on the edge of the calyx segments (Figure 6e). This feature is one of the distinguishing characters that distinguishes Barrelier's Speedwell from closely related species *V. spicata* [8]. The NG trichomes are arranged with similar density on the stem and leaves (Table 3). This type of NG trichomes is also found in *V. saturejoides* Vis. ssp. *saturejoides* [21], *V. officinalis* [30], *V. austriaca* ssp. *jacquinii* [31], *V. persica* Poir. [34], and *Veronica* sp. [35].

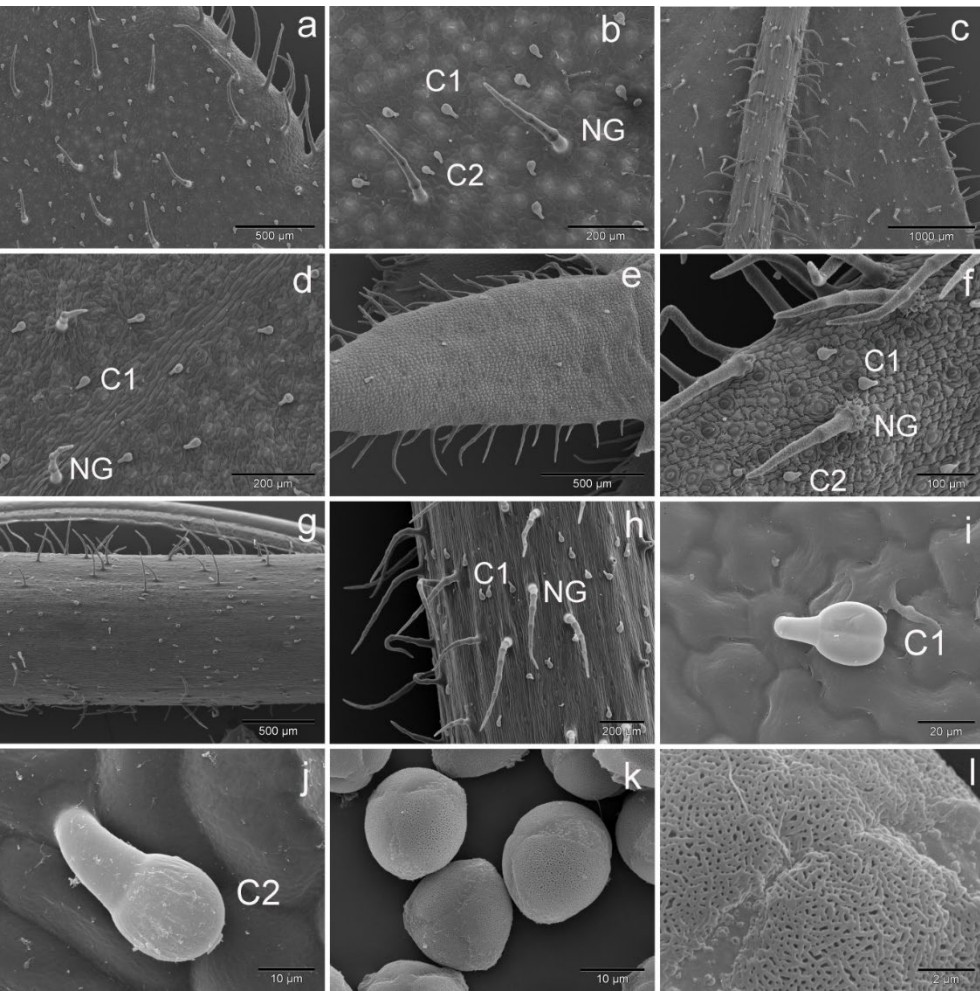

**Figure 6.** SEM micrographs of Barrelier's Speedwell with different types and distribution of trichomes (**a**–**j**) on the adaxial (**a**,**b**) and abaxial (**c**,**d**) leaf surface, calyx (**e**,**f**), and stem (**g**,**h**). Non-glandular trichomes (NG), subtype 1 (C1) and subtype 2 (C2) of capitate trichomes. SEM micrographs of pollen (**k**,**l**) after critical point drying with appearance of exine surface (**l**).

The glandular trichomes observed in Barrelier's Speedwell belong to the group of capitate trichomes. In general, the capitate trichomes and other types of glandular trichomes play an important role in plants as a site of synthesis and storage of VCs. VCs are one of the most important specialized metabolites in plants and possess considerable biological activity such as anti-inflammatory and antimicrobial activity. Two subtypes of capitate trichomes can be distinguished in Barrelier's Speedwell. The first subtype (C1) consists of a single lower cell forming a stalk and two upper cells forming the head (Figure 6i). The C1 trichomes are observed in all investigated plant parts. They are more densely arranged on the leaves and stem than on the calyx (Table 3). The same subtype of capitate trichomes was also observed in *Veronica beccabunga* L. [36], *V. saturejoides* ssp. *saturejoides* [21], *V. officinalis* [30], and *V. austriaca* ssp. *jacquinii* [31].

**Table 3.** Occurrence and frequency of trichomes in Barrelier's Speedwell. NG = non-glandular trichomes, C1 = subtype 1 of capitate trichomes, C2 = subtype 2 of capitate trichomes. Trichomes frequency: − trichomes are completely missing, ± trichomes are present in small numbers, + trichomes are present in moderate numbers, ++ trichomes are present in great numbers.

| Locality | Trichomes | Leaf | | Calyx | Stem |
|---|---|---|---|---|---|
| | | Adaxial | Abaxial | | |
| Lič Polje | NG | + | + | + | +/++ |
| | capitate C1 | + | + | ± | + |
| | capitate C2 | ± | ± | ± | ± |
| Gornja Kemenica | NG | + | + | + | + |
| | capitate C1 | + | + | + | + |
| | capitate C2 | ± | +/± | ± | +/± |
| Velebit Mt | NG | + | + | + | + |
| | capitate C1 | + | + | ±/+ | + |
| | capitate C2 | ± | +/± | −/± | +/± |
| Dinara Mt | NG | + | + | + | + |
| | capitate C1 | + | + | ± | + |
| | capitate C2 | ± | ± | ± | ± |
| Kamešnica Mt | NG | + | + | + | + |
| | capitate C1 | + | + | + | + |
| | capitate C2 | −/± | ± | ± | ± |

The second subtype of capitate trichomes (C2) consists of a single lower cell forming a stalk and upper cell forming the elliptical head (Figure 6j). C2 trichomes are also present on all investigated plant parts of Barrelier's Speedwell but with significant lower frequency than C1 trichomes (Table 3). This is the first report of the presence of C2 trichomes in *Veronica* species. Trichomes comparable to the C1 subtype of capitate trichomes have been detected in some other families such as Lamiaceae [37,38], and Geraniaceae [39,40].

The pollen grains of Barrelier's Speedwell are single (i.e., monad pollen), isopolar and radially symmetrical. The polar view shows circular-triangular shape with visible ends of three apertures (Figure 6k). In contrast, the equatorial outline is elliptical. The pollen grains have three apertures or colpi (tricolpate pollen) located in the equatorial pollen belt (zonocolpate pollen). Based on these traits it can be described as 3-zonocolpate pollen. The colpi are simple, long, rather broad, with acute apices and distinct, more or less straight margins. The membranes of the aperture are sporadically covered with irregularly arranged granular-verrucae creations. The mesocolpium is significantly larger than the apocolpium. Ornamentation of the exine of pollen grains is moderately reticulate. Reticulum meshes are unequal (heterobrochate type) and the lumina vary greatly in size. The lumina are usually narrower than muri or sometimes the same width as the muri. The shape of lumina is irregular and they have obtuse angles (Figure 6l).

According to Kremp 1965 [41], Barrelier's Speedwell has small pollen (10–25 µm) with a polar axis of $16.9 \pm 1.1$ µm, and an equatorial diameter of $19.4 \pm 1.1$ µm. The minimum and maximum values for the polar axis lengths are 15.3 and 18.8 µm, respectively. On the other hand, minimum and maximum values for the equatorial diameter are 17.5 and 21.2 µm, respectively. According to the P/E ratio (0.87), the pollen of *V. barrelieri* has an oblate-spheroidal shape. The only known study of Barrelier's Speedwell was provided in Ukraine and showed that polar axis was 21.3–23.9 µm [42], while the equatorial diameter was the same, 21.3–23.9 µm [42].

## 4. Conclusions

The free volatile compounds were isolated and analyzed using GC-MS in above ground plant parts of Barrelier's Speedwell collected from five populations in Croatia. The free volatile compounds were characterized by a high relative content of phytol (from 30.08 to 40.32%) and hexadecanoic acid (from 8.75 to 12.48%). Two subtypes of capitate trichomes (subtype 1 consisting of a single stalk cell and two head cells; subtype 2 consisting of a single stalk cell and one elliptical head cell). These results indicate that Barrelier's Speedwell may also be a source of biologically active compounds for human use and for neighboring plants if used in horticulture. Further research will focus on the biological activity of the extracts obtained and on determination of other biologicaly active compounds such as flavonoids and iridoid glycosides. Therefore, it is important to create conditions for high quality horticultural development of this decorative species, which may be threatened by climate changes.

**Author Contributions:** Conceptualization, D.K.; Methodology, V.D. and E.S.; Software, M.N.; Validation, V.D. and E.S.; Formal Analysis, V.D., E.S. and D.K.; Investigation, E.S., D.K., V.D., M.N., M.R. and S.S.; Data Curation, V.D. and E.S.; Writing—Original Draft Preparation, D.K. and V.D.; Writing—Review and Editing, D.K. and V.D.; Visualization, D.K. and V.D.; Supervision, V.D.; Funding Acquisition, V.D. and S.S. All authors have read and agreed to the published version of the manuscript.

**Funding:** This research was supported by the project "Croatian Veronica species: Phytotaxonomy and Biological Activity", CROVeS-PhyBA, funded by the Croatian Science Foundation. Project number IP-2020-02-8425.

**Data Availability Statement:** The samples are available from the authors on request.

**Conflicts of Interest:** The authors declare no conflict of interest. The founding sponsors had no role in the collection, analyses, or interpretation of data, and in the writing of the manuscript.

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
