# Peer review of "Barrelier’s Speedwell (Veronica barrelieri Schott ex Roem. et Schult., Plantaginaceae)—Potential of Free Volatile Compounds for Horticulture"

_horticulturae, doi:10.3390/horticulturae8090777_

Round 1

Reviewer 1 Report

Dear Editor, 

The article entitled “Veronica barrelieri Schott ex Roem. et Schult. (Plantaginaceae) – Biological Potential for Horticulture” is well-prepared and interesting. However, some points must be improved and clarified prior to acceptance for publication, including:

1. Title: Do the author have the English/common name of Veronica barrelieri Schott ex Roem. et Schult. (Plantaginaceae)? In my view, using English name will be more readable for the readers. 

2. Title: Using subtitle of “Biological Potential for Horticulture” is too broad. In fact, the authors only analyzed the GC-MS and morphological profile. Please make the title more specific representing the content of the manuscript.

3. Introduction: There are several previous publications regarding the GC-MS profile of Veronica sp. What is the novelty and the importance of this study?

4. Please mention explicitly the problem statement/research question and the purpose of the study.

5. Section 2.1: Which part of plant that actually used in in this study? Flower. leaves, stem or all parts of the plant? Please mention specifically. How old the plant (or the flower if the authors only used the flower). Did the authors do sortation and/or pre-treatment prior to storage?

6. Section 2.2: The authors can put the picture of the essential oil (distillate) to give more insight to the readers regarding the appearance of the Veronica barrelieri Schott extract   

7. Sub section 2.4: What is the purpose of conducting PCA analysis?

8. Discussion: volatile compounds are strongly related to aroma profile. The authors can expand the study to discuss the odor description of each compound as well as the threshold of the compounds.

9. Figure 3: Please consider to combine loading plot and score plot in one graph, so the readers can be easier to interpret the data. 

10. Discussion: It is suggested to first discuss the morphological characteristic, and then followed by the GC-MS. It will be good if the authors can combine Figure 1 and Figure 4. So, it will be clear the microstructural picture represents which part of the plant

11. Conclusions: please provide the limitation of this study and the recommendation for further studies. 

Author Response

Dear Reviewer 1,

Thank you very much, for the notes and comments. Following are the responses:

  1. Title: Do the author have the English/common name of Veronica barrelieri Schott ex Roem. et Schult. (Plantaginaceae)? In my view, using English name will be more readable for the readers. 

Reply to Reviewer: Barrelier's Speedwell.

The English species name will be applied in the title and in some parts of the article.

  1. Title: Using subtitle of “Biological Potential for Horticulture” is too broad. In fact, the authors only analyzed the GC-MS and morphological profile. Please make the title more specific representing the content of the manuscript.

Reply to Reviewer: Barrelier's Speedwell (Veronica barrelieri Schott ex Roem. et Schult.,

Plantaginaceae) – Potential of Free Volatile Compounds for Horticulture

  1. Introduction: There are several previous publications regarding the GC-MS profile of Veronica sp. What is the novelty and the importance of this study?

Reply to Reviewer: These free volatile compounds are part of the specialized metabolites and are formed in response to genetic and environmental factors. Previously published articles about GC-MS profile belong to other, mostly wider spread weed species of the genus Veronica. On the other hand, Barrelier's Speedwell possess significant horticultural traits. They allow this decorative species easy horticulture with little water, which is very important given the climatic conditions today and in an even more uncertain future. Also, closely related species Veronica spicata is used as expectorant for cough and throat rinsing in traditional medicine of Balcanic people (Popović et al., 2014). If GC-MS profile and contents of other biological active compounds (should be determined through future research) in V. barrelieri and V. spicata are similar there is a real possibility to use V. barrelieri as a substitute for V. spicata. So, the determination of GC-MS profile in V. barrelieri is a first step in evaluation of using this species in horticultural and maybe medicinal purposes.

  1. Please mention explicitly the problem statement/research question and the purpose of the study.

Reply to Reviewer: The aim of the research is to learn the anatomical structure of the glandular hairs where free volatile compounds are formed, in order to discover the biological potential for the horticultural development of this ornamental species.

  1. Section 2.1: Which part of plant that actually used in in this study? Flower. leaves, stem or all parts of the plant? Please mention specifically. How old the plant (or the flower if the authors only used the flower). Did the authors do sortation and/or pre-treatment prior to storage?

Reply to Reviewer: The above-ground part of the plant during the flowering period was used for the study, i.e. flowers, leaves and stems. The plant material was air-dried and stored as described in section Material and Methods.

  1. Section 2.2: The authors can put the picture of the essential oil (distillate) to give more insight to the readers regarding the appearance of the Veronica barrelieri Schott extract   

Reply to Reviewer: Thanks for pointing this out, but Veronica barrelieri extracts are stored in dark bottles, and if the samples are transferred to light bottles, there is a possibility that some of the distillate will be lost. As we have already mentioned, the resulting distillate is small in amount and will be used for further biological research.

  1. Sub section 2.4: What is the purpose of conducting PCA analysis?

Reply to Reviewer: The purpose of conducting PCA analysis is to objectively determine which populations of Veronica barrelieri are more similar to each other in terms of GC-MS profile.

  1. Discussion: volatile compounds are strongly related to aroma profile. The authors can expand the study to discuss the odor description of each compound as well as the threshold of the compounds.

Reply to Reviewer: As we mentioned before, the future study will investigate the biological activity of the extracts. The biological activity of the total extracts will be compared with each of the main components in the composition of the distillate, and then the description of the odour of each compound will be discussed, as well as the threshold value of the compounds.

  1. Figure 3: Please consider to combine loading plot and score plot in one graph, so the readers can be easier to interpret the data.

Reply to Reviewer : The presented way is common and maybe it is be better that Figure 3 stay unchanged. We attach here the Biplot for viewing. In our opinion keeping the PCA analyses Loadings and PC Score plot is better.

  1. Discussion: It is suggested to first discuss the morphological characteristic, and then followed by the GC-MS. It will be good if the authors can combine Figure 1 and Figure 4. So, it will be clear the microstructural picture represents which part of the plant.

Reply to Reviewer : The main purpose of this study is to obtain GC-MS profile of Veronica barrelieri and maybe it is more adequate to present main result, that is GC-MS profile. After that, the place of origin and storage of essential oil (glandular trichomes) is presented. Furthermore, Figure 1 describe appearance of V. barrelieri in general, not plant parts. Because of that, it is not adequate to connect plant parts with micrographs.

  1. Conclusions: please provide the limitation of this study and the recommendation for further studies. 

Reply to Reviewer: Further research will focus on the biological activity of the extracts obtained and on determination other biological active compounds such as flavonoids and iridoid glycosides. Therefore, it is important to create conditions for high quality horticultural development of this decorative species, which may be threatened by climate changes.

Reviewer 2 Report

All the research carried out in plant species that are low water demandant is quite interesting due to the climate reality and future situation. Some points of the manuscript must be checked and explained before reconsideration for publication:

Introduction: The potential biological activity claimed by the authors of the volatile compounds of plants should be developed and included in the text of the introduction to justify the research.

Line 98: The fifty grams of the plant should be described carefully because the amount of stem is going to influence the terpenes found. Please, specify as much as possible.

Line 102: The main results of the paper are obtained from the analysis of volatile compounds, therefore is necessary to detail all the procedure followed from Nazlić et al., not only citing this study but writing the procedure carried out. Also, the identification must be more detailed. It is not clear what compounds were identified with authentic reference standards and what compounds LRI coincide with the literature.

Line 125: It is not clear what the authors meant with “the yield is 0.07%”. Please specify more explanation.

Line 172: Please, be careful when including vitamin E as a volatile compound.

Line 346: This sentence is not a conclusion, please rephrase it.

Table 2: What does “co-injection with reference compounds” mean?

Author Response

Dear Reviewer 2,

Thank you very much for the notes and comments. Following are the responses:

All the research carried out in plant species that are low water demandant is quite interesting due to the climate reality and future situation. Some points of the manuscript must be checked and explained before reconsideration for publication:

Introduction: The potential biological activity claimed by the authors of the volatile compounds of plants should be developed and included in the text of the introduction to justify the research.

Reply to Reviewer: The suggestion was accepted and other sentences were added: In addition to the effect of all components, biological activity can also be influenced by individual components in the composition of the isolate. Thus, in the composition of the studied isolate of V. barellieri, the main component is phytol, a diterpene alcohol from chlorophyll known for its medicinal properties. Phytol has antimicrobial, antioxidant and anti-inflammatory activity. Several studies have demonstrated the anti-cancer effects of plant extracts with phytol as the main ingredient. The effects of phytol were also confirmed on apoptosis in hepatocellular carcinoma cells and the cytotoxic effect of phytol on cell lines in vitro

Line 98: The fifty grams of the plant should be described carefully because the amount of stem is going to influence the terpenes found. Please, specify as much as possible.

Reply to Reviewer: We used the aerial parts of the plant for the extraction: flower, leaf and stem. We did not separate the stem because, as the micromorphological pictures show, there are C1 and C2 capitate trichomes on the stem, which are the synthesis sites of the volatile compounds. Further research of GC-MS profile and other biological active compounds of V. barrelieri could be done on separate plant parts (leaves, flower, stem). On that way, the mentioned influence of stem on terpenes quantity will be avoided.

Line 102: The main results of the paper are obtained from the analysis of volatile compounds, therefore is necessary to detail all the procedure followed from Nazlić et al., not only citing this study but writing the procedure carried out. Also, the identification must be more detailed. It is not clear what compounds were identified with authentic reference standards and what compounds LRI coincide with the literature.

Reply to Reviewer: The procedure is explained in much more details.

Line 125: It is not clear what the authors meant with “the yield is 0.07%”. Please specify more explanation.

Reply to Reviewer: It means that the obtained quantity of essential oil is 0.07 percent of weight of plant material which was subjected to hydrodistillation. That means, if you have 50 g of dried plant material, and the yield is 0.07%, weight of the volatile compounds extracted is 0.035 g or 35 mg.

Line 172: Please, be careful when including vitamin E as a volatile compound.

Reply to Reviewer: Thank you very much for your comment, we deleted vitamin E in the article.

Line 346: This sentence is not a conclusion, please rephrase it.

Reply to Reviewer: We rephrased it. “The free volatile compounds were isolated and analyzed using GC-MS in above ground plant parts of Barrelier's Speedwell collected from five populations in Croatia. The free volatile compounds were characterized by a high relative content of phytol (from 30.08 to 40.32%) and hexadecanoic acid (from 8.75 to 12.48%). Two subtypes of capitate trichomes (subtype 1 consisting of a single stalk cell and two head cells; subtype 2 consisting of a single stalk cell and one elliptical head cell). These results indicate that Barrelier's Speedwell may also be a source of biologically active compounds for human use and for neighboring plants if used in horticulture. Further research will focus on the biological activity of the extracts obtained and on determination of other biologicaly active compunds such as flavonoids and iridoid glycosides. Therefore, it is important to create conditions for high quality horticultural development of this decorative species, which may be threatened by climate changes.

Table 2: What does “co-injection with reference compounds” mean?

Reply to Reviewer: We injected standards (marked with an asterisk in the table 2) for specific compounds into the GC-MS under the same operating conditions as the isolates obtained so the identification of the compounds is more accurate.

Reviewer 3 Report

This is an interesting paper, but you may improve this article to publish in this journal. Otherwise, I have a lot of recommendations to increase the quality of your paper. Be careful with the writing and mistakes.

Line 22. When you write an acronym, you must write in capitals the letters that you use to build it, so, you must write “Scanning Electronic Microscopy”. This makes the reading easier for potential readers. You must fix this tiny mistake in the whole manuscript and you must look for it.

Lines 20, 75, 80 and 99-100. With the same idea you must write “Volatile Compounds”.

Line 80. Rephrase this sentence. “VCs” is redundant best write as follows: “To the best of our knowledge there is no information about free Volatile Compounds of V. barrelieri.”.

Line 88. This is a mistake. You must write five instead of four localities. In the whole paper you have studied five localities.

Line 93. You must write “Scanning Electron Microscope”. Please, write in capitals the letters which form the acronym. This makes the reading easier.

Line 101. You must write in capitals the letters which form the acronym: “Gas Chromatography and Mass Spectrometry”. This is to make the reading easier.

Line 106. You must write the author of the species Veronica barrelieri.

In Table 1 you must write the degrees of the latitude and longitude of the fourth locality which is Dinara Mt.

Lines 120-121. You must write “Principal Component Analysis” in capitals because these are the letters which form the acronym PCA.

In material and methods, you must show us a map of the localities you have studied, and if it is possible you must draw the distribution of the species. This is very useful to best understand the paper.

Line 148. I prefer to write “acids” because you have described some of them.

Line 153. You must write “EOs” instead of “Eos”. As well you must write “Essential Oils” in capitals for the reasons formerly explained.

Line 158. “21.0” is a percentage but you must write the symbol “%” just after the number.

Line 164. You must write the author of V. officinalis because in a botanical journal the first time that you write a species it is compulsory to write its authors.

Line 166. You must write the authors of the species V. austriaca ssp. jacquinii the very first time that you write a species, not the last time. You have written its authors in line 244. Please, fix this mistake.

Line 183. You must finish the sentence with a point. As well you must write the author of the species.

Line 213. The word Polje must be written in capitals as you have written it in the Table 1.

Line 262. The families must end with the letters “aceae”, so, you must write “Geraniaceae”.

Line 296. You must write the word Polje must be written in capitals as you have written it in the Table 1.

The numbers 292 and 297 must be on the right, not on the left of the article in the lines 292 and 297.

Otherwise, the authors adequately developed the Introduction, presenting the problems but you must write explicitly the objectives of this paper.

The methods are adequate.

The Discussion is well developed and the data presented are correctly compared with other papers.

The authors are to be congratulated for the results obtained in this article.

Author Response

This is an interesting paper, but you may improve this article to publish in this journal. Otherwise, I have a lot of recommendations to increase the quality of your paper. Be careful with the writing and mistakes.

Line 22. When you write an acronym, you must write in capitals the letters that you use to build it, so, you must write “Scanning Electronic Microscopy”. This makes the reading easier for potential readers. You must fix this tiny mistake in the whole manuscript and you must look for it.

Reply to Reviewer: The suggestion was accepted.

Lines 20, 75, 80 and 99-100. With the same idea you must write “Volatile Compounds”.

Reply to Reviewer: The suggestion was accepted.

Line 80. Rephrase this sentence. “VCs” is redundant best write as follows: “To the best of our knowledge there is no information about free Volatile Compounds of Vbarrelieri.”.

Reply to Reviewer: The suggestion was accepted.

Line 88. This is a mistake. You must write five instead of four localities. In the whole paper you have studied five localities.

Reply to Reviewer: The suggestion was accepted.

Line 93. You must write “Scanning Electron Microscope”. Please, write in capitals the letters which form the acronym. This makes the reading easier.

Reply to Reviewer: The suggestion was accepted.

Line 101. You must write in capitals the letters which form the acronym: “Gas Chromatography and Mass Spectrometry”. This is to make the reading easier.

Line 106. You must write the author of the species Veronica barrelieri.

Reply to Reviewer: According to the suggestion of the Reviewer 1 'Veronica barrelieri‘ was changed into ‘Barrelier's Speedwell'.

In Table 1 you must write the degrees of the latitude and longitude of the fourth locality which is Dinara Mt.

Reply to Reviewer: It was done.

Lines 120-121. You must write “Principal Component Analysis” in capitals because these are the letters which form the acronym PCA.

Reply to Reviewer: It was done.

In material and methods, you must show us a map of the localities you have studied, and if it is possible you must draw the distribution of the species. This is very useful to best understand the paper.

Reply to Reviewer: The map was added.

Line 148. I prefer to write “acids” because you have described some of them.

Reply to Reviewer: The suggestion was accepted.

Line 153. You must write “EOs” instead of “Eos”. As well you must write “Essential Oils” in capitals for the reasons formerly explained.

Reply to Reviewer: The suggestion was accepted.

Line 158. “21.0” is a percentage but you must write the symbol “%” just after the number.

Reply to Reviewer: The symbol was added

Line 164. You must write the author of Vofficinalis because in a botanical journal the first time that you write a species it is compulsory to write its authors.

Reply to Reviewer: The suggestion was accepted.

Line 166. You must write the authors of the species Vaustriaca ssp. jacquinii the very first time that you write a species, not the last time. You have written its authors in line 244. Please, fix this mistake.

Reply to Reviewer: The mistake has been corrected.

Line 183. You must finish the sentence with a point. As well you must write the author of the species.

Reply to Reviewer: The suggestion was accepted.

Line 213. The word Polje must be written in capitals as you have written it in the Table 1.

Reply to Reviewer: The mistake has been corrected .

Line 262. The families must end with the letters “aceae”, so, you must write “Geraniaceae”.

Reply to Reviewer: The mistake has been corrected .

Line 296. You must write the word Polje must be written in capitals as you have written it in the Table 1.

Reply to Reviewer: The mistake has been corrected .

The numbers 292 and 297 must be on the right, not on the left of the article in the lines 292 and 297.

Reply to Reviewer: The mistake has been corrected.

Otherwise, the authors adequately developed the Introduction, presenting the problems but you must write explicitly the objectives of this paper.

Reply to Reviewer:  This has been added at he ends of the Introduction: The aim of this paper is to obtain GC-MS profile of free VCs of Barrelier's Speedwell and to learn the anatomical structure of the glandular hairs where VCs are formed, in order to discover the biological potential for the horticultural development of this ornamental species.

The methods are adequate.

The Discussion is well developed and the data presented are correctly compared with other papers.

The authors are to be congratulated for the results obtained in this article.

Reviewer 4 Report

Interesting results for possible environmentally friendly applications in agriculture and sustainable nurseries.

The contribution expands the knowledge of the active metabolites present in the species of the genus Veronica

Author Response

Dear Reviewer 4,

Thank you very much for your comments

Round 2

Reviewer 2 Report

In reference to the co-injection named by the authors, if they refer that they injected the standards afterwards and checked that the LRI of the authentic compounds was the same that the one observed in the compounds marked with an asterisk, the term co-injection is incorrect. Did the authors after the injection of the samples, injected again the sanples with the standards added? Please, this deserves a properly explanation.   

In line 129, please chek if the sentence is correct. authentic samples? or authentic standards?

Author Response

Dear Reviewer,

Thank you!
